# Visual Tea Leaf Disease Recognition Using a Convolutional Neural Network Model

**Jing Chen** [1,†] 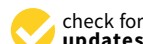, **Qi Liu** [2,†] **and Lingwang Gao** [1,*]

1   College of Plant Protection, China Agricultural University, Beijing 100193, China; cj-yx2004@163.com
2   College of Agronomy, Xinjiang Agricultural University, Key Laboratory of the Pest Monitoring and Safety Control of Crops and Forests, Urumqi 830052, China; stanton67@126.com
*   Correspondence: lwgao@cau.edu.cn
†   Both authors contributed equally.

**Abstract:** The rapid, recent development of image recognition technologies has led to the widespread use of convolutional neural networks (CNNs) in automated image classification and in the recognition of plant diseases. Aims: The aim of the present study was to develop a deep CNNs to identify tea plant disease types from leaf images. Materials: A CNNs model named LeafNet was developed with different sized feature extractor filters that automatically extract the features of tea plant diseases from images. DSIFT (dense scale-invariant feature transform) features are also extracted and used to construct a bag of visual words (BOVW) model that is then used to classify diseases via support vector machine(SVM) and multi-layer perceptron(MLP) classifiers. The performance of the three classifiers in disease recognition were then individually evaluated. Results: The LeafNet algorithm identified tea leaf diseases most accurately, with an average classification accuracy of 90.16%, while that of the SVM algorithm was 60.62% and that of the MLP algorithm was 70.77%. Conclusions: The LeafNet was clearly superior in the recognition of tea leaf diseases compared to the MLP and SVM algorithms. Consequently, the LeafNet can be used in future applications to improve the efficiency and accuracy of disease diagnoses in tea plants.

**Keywords:** convolutional neural networks; DSIFT; SVM; MLP; tea disease; classification

## 1. Introduction

Tea plantation and production areas comprise a total of 2.87 million hectares among the 17 provinces of China. Moreover, the total output of tea plant production exceeded 2.4 billion tons in 2016 [1]. Tea plants tolerate elevated levels of heat and shade, and thus, areas where tea plantations are typically found are characterized by warm climates and abundant rainfall. However, these regions are also very conducive to the growth and reproduction of diseases that have severely decreased tea quality with the gradual increase in tea production. Consequently, tea diseases are a limiting factor hindering robust tea production. Plant disease diagnosis is typically based on the characteristic appearances of diseases. However, trained tea plant pathologists are scarce, and limitations in the background knowledge of tea growers leads to an inability to identify disease events in a timely and effective manner. Therefore, development and implementation of a diagnostic framework for tea plant diseases would help ensure the accurate and timely identification of tea plant diseases by agricultural producers. Such improvements would lead to better control methods that would economically and effectively restore losses due to diseases. Moreover, these advancements would help ensure higher tea qualities while reducing costs of labor and agricultural production, and importantly, thereby improving yields and the sustainable development of tea production.

The current methods for diagnosing plant diseases primarily include microscopic identification in addition to molecular biological and spectroscopic techniques. Microscopic identification is time consuming and can be subjective, where even experienced plant pathologists may incorrectly diagnose diseases. Molecular biological and spectroscopic identification are more accurate but are labor intensive and require specialized and expensive instrumentation.

The rapid development of intelligent agriculture and precision agriculture in recent years has led to the widespread use of computer image processing technologies to solve diverse problems within agricultural sciences. For example, these technologies have been used to estimate plant nutrient content [2–4], classify plant species [5], and identify plant diseases [6,7]. In particular, deep neural network and genetic algorithms have been used in combination to estimate nitrogen content in wheat leaves [2–4], which represents a considerable improvement over other existing methods.

Artificial neural networks (ANNs) and Support Vector Machines (SVMs) have been used in this capacity. ANNs reflect biological neural networks and autonomously learn, progressively improving their knowledge and capacities [8]. The multi-layer perceptron (MLP) model is a multi-layer feedforward artificial neural network model that exhibits superior performance when analyzing nonlinear systems. MLPs typically comprise the input, hidden, and output layers, where each layer contains several neurons, the sigmoidal linear activation functions was used to map the sum of the weighted inputs to the output of a neuron in the hidden layer [9].

SVMs are an effective type of classification algorithm that has been widely used for many pattern recognition tasks [10], including machine learning methods based on statistical learning theory [11]. The primary goal of SVMs is to identify a separating plane to evaluate different class memberships. SVMs were initially used to classify two-class problems in the analysis of linear separable cases. In the event of linear inseparability, nonlinear mapping algorithms can be used to transform linearly inseparable samples of low-dimensional input space into high-dimensional feature space in order to render them linearly separable. The technique is based on the structural risk minimization theory that informs the construction of an optimal hyperplane in feature space, such that the learner is globally optimized and the expectation in the entire sample space meets a certain upper bound with a certain probability. These two methods require smaller sample sizes and an appropriate train rule, which have led to their widespread use in image classification and recognition.

Convolutional neural networks (CNNs) were developed in the 1980s and are a type of deep neural network which were used to recognize handwritten digits [12]. CNNs could learn to extract features from images by themselves through stacked layers of convolutional filters. Typical CNNs are hierarchical neural networks that are primarily composed of multiple convolution layers, a pooling layer, and a full connection layer. Local receptive fields, weight sharing, and spatial sub-sampling are the primary hierarchical aspects within the networks. These attributes result in a high invariance of CNNs for translation, scaling, shifting, or other forms of deformation [13]. Moreover, CNNs directly take the image as a network input, thus avoiding the extraction of complex features and the need for data reconstruction, as in traditional image recognition algorithms. Meanwhile, the high recognition accuracy of CNNs leads to wide implementation in fields related to computer vision, where development is occurring rapidly [14].

The rapid development of computer vision technology in recent years has led to increased usage of computational image processing and recognition methods to identify diseases. The most widely used current method relies on extracting global features including color features [15], shape features [16], texture features [17], or some combination of the above features [18–22]. The local features of the disease spots are then processed using various algorithms including local feature SIFT (scale-invariant feature transform), SURF (speeded-up robust features), HOG (histogram of oriented gradient), DSIFT (dense scale-invariant feature transform), and PHOW (pyramid histograms of visual words) [23–25]. Lastly, the extracted feature parameters are used in various classifiers including ANNs [26,27] and SVMs [28,29]. A significant drawback of these methods is the need to artificially extract features in advance. In contrast, CNNs learn data characteristics from convolution operations, which is better

suited for pattern recognition of images. Consequently, CNNs have been used to detect and diagnose plant diseases [30–34].

Following these previous studies, CNNs were developed and adopted here to improve the diagnosis and classification of tea diseases. In addition, the classification performances of traditional machine learning algorithms were evaluated relative to manual classifications and the proposed CNNs. Among the former, the HOG, SURF, and PHOW algorithms did not exhibit better invariance towards image rotation and scaling than SIFT, while the SIFT algorithm performed reliably with affine transformations, viewing angle variation, and noise. Moreover, the SIFT algorithm exhibited strong scalability, that when combined with other algorithms could be used as a highly optimized algorithm. Consequently, SIFT was used here as a feature descriptor in a traditional machine learning algorithm. Although SIFT features can describe images, each SIFT represents a 128-dimensional vector, and images contain hundreds or thousands of SIFT features, thereby leading to very computationally intensive operations. To greatly reduce computational processing, a bag of visual words (BOVW) model was constructed based on these vectors, wherein each image was represented by a numerical vector.

## 2. Materials and Methods

To identify the optimal strategy for identifying tea leaf diseases from images, disease classification was conducted with CNNs and compared against classifications with SVM and multi-layer perceptron (MLP) algorithms. The dense scale-invariant feature transform (DSIFT)-based bag of visual words (BOVW) model was used to obtain image features for the latter two classifiers.

### 2.1. Disease Dataset

Images showing tea leaf diseases were all captured using a Cannon PowerShot G12 camera in the natural environments of Chibi and Yichang within the Hubei province of China. The images were taken ~20 cm directly above the leaves and captured in the auto-focus mode at a resolution of 4000 × 3000 pixels. A total of 3810 tea leaf images were used that showed symptoms for seven different diseases, as identified by phytopathologists (Figure 1). The identification criteria used for the tea tree diseases were based on previously described identification schemes [35,36]. All images in the present manuscript were resized to 256 × 256 pixels. In order to improve the classifier's generalization ability, we increased the size of the dataset, which is more advantageous to the training of the network. Three different methods were used to alter the image input and improve classification (Figure 2). Finally, there are 7905 images in the database. Table 1 shows the number of images for every class used as training, validation and testing datasets for the disease classification model.

The disease classification datasets that were used in these analyses are shown in Table 1. The 80/20 ratio of training/test data is the most commonly used ratio in neural network applications. In addition, a 10% subset of the test dataset was used to validate the dataset.

**Table 1.** The image dataset comprising of seven different diseases used in this study.

| Class | Number of Images from the Dataset Used for Training | Number of Images from the Dataset Used for Validation | Number of Images from the Dataset Used for Testing |
|---|---|---|---|
| (1) White spot | 941 | 118 | 117 |
| (2) Bird's eye spot | 955 | 120 | 119 |
| (3) Red leaf spot | 890 | 111 | 111 |
| (4) Gray blight | 893 | 112 | 111 |
| (5) Anthracnose | 880 | 110 | 110 |
| (6) Brown blight | 920 | 115 | 115 |
| (7) Algal leaf spot | 846 | 106 | 105 |
| Total | 6325 | 792 | 788 |

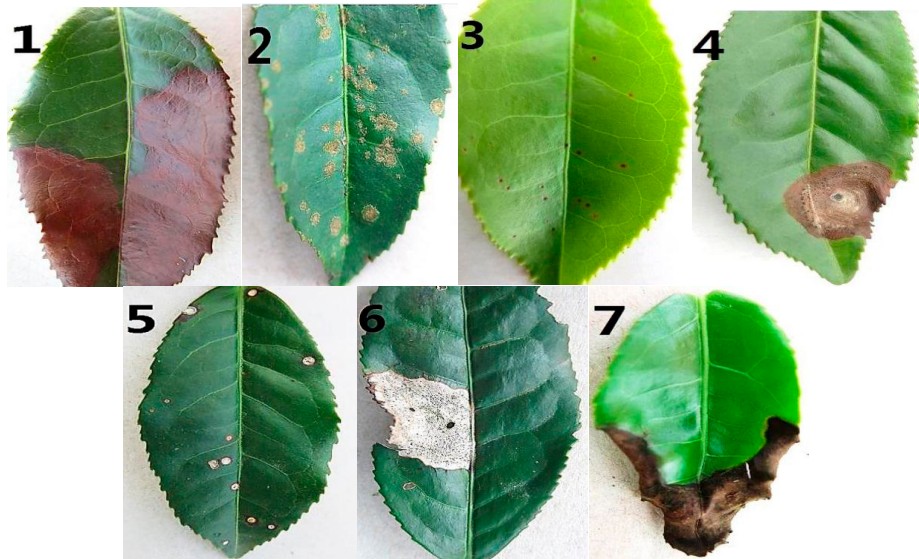

**Figure 1.** Typical example images of tea leaves used in the present analysis. (**1**) Red leaf spot caused by *Phyllosticta theicola Petch*; (**2**) Algal leaf spot caused by *Cephaleuros virescens Kunze*; (**3**) Bird's eye spot caused by *Cercospora theae Bredde Haan*; (**4**) Gray blight caused by *Pestalotiopsis theae (Sawada) Steyaert*; (**5**) White spot caused by *Phyllosticta theaefolia Hara*; (**6**) Anthracnose caused by *Gloeosporium theae-sinensis Miyake*; (**7**) Brown blight caused by *Colletotrichum camelliae Massee*.

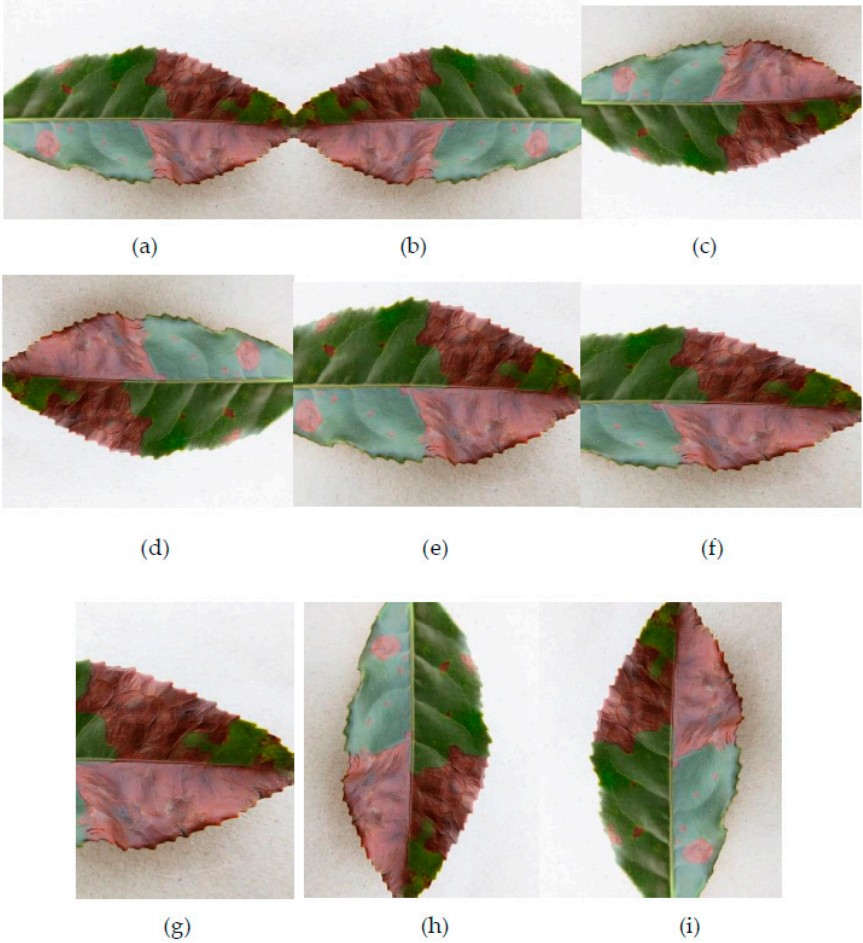

**Figure 2.** Examples of data augmentation used for red leaf spot images. (**a**) initial; (**b**) flip horizontal; (**c**) flip vertical; (**d**) rotated 180°; (**e–g**) randomly cropped; (**h**) right-rotate 90°; (**i**) left-rotate 90°.

## 2.2. Performance Measurements

The accuracy and mean class accuracy (MCA) indices were used to evaluate algorithm performances, as previously described [37]. $CCR_k$ is first defined as the correct classification rate for class $k$, as determined by Equation (1):

$$CCR_k = \frac{C_k}{N_k} \tag{1}$$

where $C_k$ is the number of correct identifications for class $k$ and $N_k$ is the total number of elements in class $k$. Classification accuracy is then defined by Equation (2):

$$\text{Accuracy} = \frac{\sum_k CCR_k \cdot N_k}{\sum_k N_k} \tag{2}$$

Lastly, MCA is determined using Equation (3):

$$\text{MCA} = \frac{1}{k} \sum_k CCR_k \tag{3}$$

## 2.3. Convolutional Neural Networks

The network architecture described here is an improvement upon the classical AlexNet model, and is termed LeafNet. The total number of parameters (weights and biases) of the entire AlexNet

network reaches upwards of 60 million, wherein the parameters of the convolution layer comprises 3.8% of the total network parameters and those of the full connection layer comprises up to 96.2% of the total. To reduce the computational complexity associated with such networks, we sought to construct a disease identification model with a relatively simple structure and small computational requirements. The network was constructed by reducing the number of convolutional layer filters and the number of nodes in the fully connected layer, thereby effectively reducing the number of network parameters requiring training and reducing the overfitting problem.

In particular, the LeafNet architecture comprises of five convolutional layers, two fully connected layers, and a final layer as the classification layer. In addition, the number of filters in the first, second, and fifth convolutional layers are designed to equal half of those used in AlexNet's filters. Further, the number of neurons in the fully connected layer are 500, 100, and seven, respectively, which differs from the number used in the standard AlexNet architecture. The entire network architecture is shown in Figure 3.

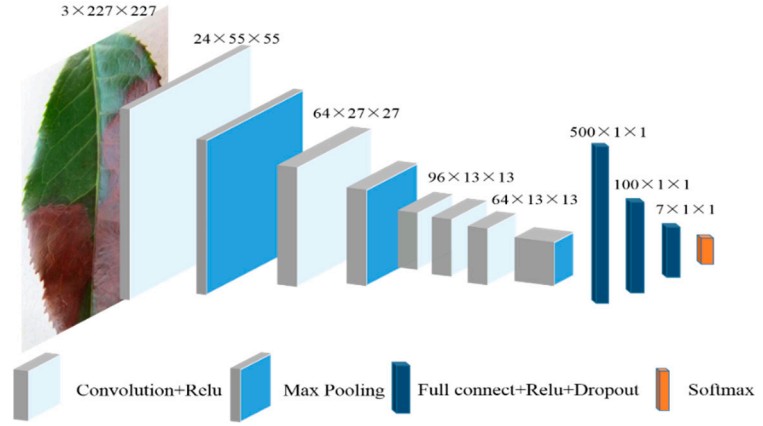

**Figure 3.** Architecture of the LeafNet that was used for disease recognition.

Input images were rescaled to $227 \times 227$ pixels and the three color channels were all processed directly by the network. The convolutional and full connection layers are defined as follows:

1. The first convolutional layer comprises of 24 filters and a kernel size of $11 \times 11$ pixels, outputting 24 feature maps with a size of $55 \times 55$ pixels. This process is followed by a rectified linear unit (ReLU) operation. ReLU is an activation function that provides a solution to vanishing gradients and exhibits more optimal error transmission than the sigmoid function. The pooling layer has a kernel size of $3 \times 3$ pixels, with a stride of 2 pixels. Local Response Normalization (LRN) after pooling can improve the generalization ability of models and accelerate the convergence speed of networks. Finally, $24 \times 27 \times 27$ feature maps are obtained.

2. The second convolutional layer comprises of 64 filters with kernel sizes of $5 \times 5$ pixels and outputs $64 \times 27 \times 27$ feature maps. As above, batch normalization and ReLU operations are also implemented. The pooling layer has a kernel of size $3 \times 3$ pixels, with a stride of 2 pixels. After pooling, local response normalization is performed, and $64 \times 13 \times 13$ feature maps are obtained.

3. The third and fourth convolutional layers both comprise of 96 filters with kernel sizes of $3 \times 3$ pixels, and output $96 \times 13 \times 13$ feature maps that are subjected to a ReLU operation.

4. The fifth convolutional layer has 64 filters with kernel sizes of $3 \times 3$ pixels and outputs $64 \times 13 \times 13$ feature maps that are subjected to a ReLU operation. The pooling layer has a kernel size of $3 \times 3$ pixels and a stride of 2 pixels. After pooling, $64 \times 6 \times 6$ feature maps are obtained.

5. The first full connection layer contains 500 neurons and is followed by a ReLU operation in addition to a dropout operation. The dropout [38] technique is an effective solution to overfitting via the training of only some of the randomly selected nodes rather than the entire network. The dropout ratio was set as 0.5, as used previously.

6. The second full connection layer contains 100 neurons and is followed by ReLU and dropout operations.

7. The last full connection layer contains seven neurons, representing the number of tea leaf disease categories. The output of the last full connection layer is then transferred to the output layer to determine the classification of the input image. The softmax activation function is then implemented, which forces the sum of the output values to equal 1.0 and limits individual outputs to values between 0–1. The softmax function is an appropriate implementation into LeafNet because it accounts for the relative magnitudes of all outputs. Layer parameters for the LeafNet are shown in Table 2.

**Table 2.** Layer parameters for the LeafNet.

| *Layer* | *Parameter* | *Activation Function* |
|---|---|---|
| Input | $227 \times 227 \times 3$ | - |
| Convolution1 (Conv1) | 24 convolution filters ($11 \times 11$), 4 stride | ReLU |
| Pooling1 (Pool1) | Max pooling ($3 \times 3$), 2 stride | - |
| Convolution2 (Conv2) | 64 convolution filters ($5 \times 5$), 1 stride | ReLU |
| Pooling2 (Pool2) | Max pooling ($3 \times 3$), 2 stride | - |
| Convolution3 (Conv3) | 96 convolution filters ($3 \times 3$), 1 stride | ReLU |
| Convolution4 (Conv4) | 96 convolution filters ($3 \times 3$), 1 stride | ReLU |
| Convolution5 (Conv5) | 64 convolution filters ($3 \times 3$), 1 stride | ReLU |
| Pooling5 (Pool5) | Max pooling ($3 \times 3$), 2 stride | - |
| Full Connect6 (fc6) | 500 nodes, 1 stride | ReLU |
| Full Connect7 (fc7) | 100 nodes, 1 stride | ReLU |
| Full Connect8 (fc8) | 7 nodes, 1 stride | ReLU |
| Output | 1 node | Softmax |

Weights for all of the layers are first initialized with random values from a Gaussian distribution. The network was trained using a stochastic gradient descent (SGD) technique, with a batch size of 16 and a momentum value of 0.9 [39]. Batch training increases the convergence rate while minimizing memory usage. The initial learning rate of all layers of the network were set to 0.1. As the network trains, the learning rate decreased according to the error reduction. Specifically, the learning rate decreases to 0.1 times the original learning rate in subsequent iterations, with the minimum learning rate threshold set to 0.0001. The number of epochs was set as 100, while the weight of decay was set to 0.0005. The method was implemented using the Matconvnet toolbox for MATLAB and training was performed on a Lenovo machine with a Core i7-3770K CPU, 8 GB of RAM, and acceleration via two NVIDIA GeForce GTX 980 GPUs.

### 2.4. Dense SIFT-Based Bag of Visual Words (BOVW) Model

Lowe [40] proposed the SIFT algorithm for extracting local features from images which is invariant to image rotation, scaling, and affine transformations. Consequently, the SIFT method is considered the most robust local invariant feature descriptor for image processing [41]. However, for the SIFT algorithm that includes feature detection and description stages, considerable calculation requirements lead to the low image processing speed. Therefore, an improvement upon the original algorithm, the dense scale-invariant feature transform (DSIFT) algorithm [42], has been developed and applied. The DSIFT algorithm applies a fixed-size rectangular window for sampling, proceeding from the left to the right of the image and from the top to the bottom, based on a specified step length. The center of the window is used as the key point and an image block comprising of 16 pixels around the center is divided into cells with $4 \times 4$ pixel sizes. Within each cell, a gradient histogram is calculated in eight

directions using the SIFT algorithm, and a feature vector of $4 \times 4 \times 8 = 128$ dimensions is obtained, thus forming the DSIFT descriptor.

The BOVW model proposed by Sivic and Zisserman [43] is widely used in machine vision due to its simplicity and computational efficiency [44]. The traditional BOVW model comprises four major steps. The first step is feature extraction and description. The DSIFT output represents the local invariant feature points for each image and were the dense SIFT descriptors in this study, which comprised of a 128 dimensional vector. The second step is the construction of a visual vocabulary by processing the dense SIFT descriptors using a K-means algorithm. Each cluster center can be thought of as a visual word in the dictionary. All visual words thus form a visual vocabulary and the size of the vocabulary is equal to the number of words. The third step is to statistically analyze the number of visual word occurrences in each image, wherein the image can be represented as a numerical vector histogram. The final step is the operation of a classifier, which was either a support vector machine (SVM) or multi-layer perceptron (MLP) algorithm in this study.

In this study, the tea disease image was represented by the DSIFT feature vector of size $1000 \times 128$, with a *k* value of 1000. The input image histogram was then sent to the SVM and MLP classification algorithms. Here, a three-layer MLP network structure is adopted with the number of nodes in the hidden layer set at 100 and the use of a sigmoid activation function. The random gradient descent method with mini-batch was used to train the MLP parameters, with a batch size of 25 and a learning rate of 1.0. The framework of the SVM and MLP algorithms based on the BOVW model is shown in Figure 4.

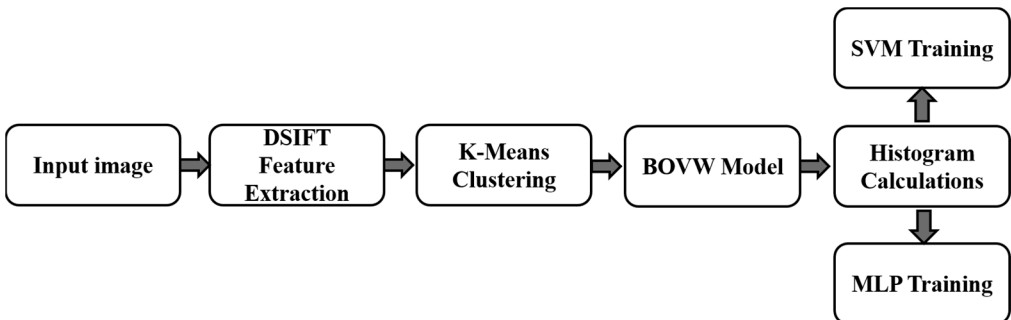

**Figure 4.** Architecture of the SVM and MLP models used for disease recognition.

## 3. Results and Discussion

In this study, the accuracy of the SVM, MLP, and CNN classifiers in determining disease states for tea leaves from images were evaluated. The results of these analyses are shown in Figure 5. Error matrices were used to evaluate the accuracy of tea leaf disease recognition classifiers (Tables 3–5). From these data, the various tea leaf disease recognition algorithms generally identified the majority of diseases correctly, although the LeafNet algorithm clearly performed better than the SVM and MLP algorithms. Among the different diseases, the Bird's eye spot disease was best distinguished by the LeafNet, which is likely due to its obvious phytopathological symptoms and ease of discernment. White spot disease was the next most accurately classified disease, while all other diseases were classified at accuracies between 84–93%. The gray blight, red leaf spot, and brown blight were classified with the least accuracy, which is due to the similarity in pathological characteristics among the three diseases. Some disease symptoms are too similar in their later stages to be distinguished, like gray blight and brown blight diseases, which both exhibit annulations during later stages. Moreover, the symptoms in the early and middle stages of these diseases are also difficult to distinguish. In addition to the above cases, the symptoms during early and middle stages of some diseases are also very similar. For example, the symptoms of white spot and bird's eye spot diseases both include reddish brown spots at early stages. In addition, both anthracnose and brown blight diseases are typified by waterlogged leaves during early disease stages, while differentiation occurs during later stages.

Some diseases can occur in tea plants throughout the year, although some diseases occur at distinct times. Consequently, the time of year when diseases are diagnosed may differ, which would affect the accuracy of disease recognition. A further complication in accurate disease diagnoses may be that tea leaves can be infected by two or more diseases. The occurrence of a disease within a leaf would likely result in physiological weakness that could lead to infection by a second disease. Thus, the above confounding factors may explain lessened accuracy in disease recognition by the tested models.

In addition, the performance of LeafNet was compared against two other methods described previously [26,28]. The accuracy of LeafNet was slightly lower than of the two aforementioned algorithms (Table 6). However, more types of diseases were used in the present study compared to those used to evaluate the accuracy of the other two models. Consequently, the method proposed here to classify tea tree diseases is clearly superior to the two other previously described algorithms.

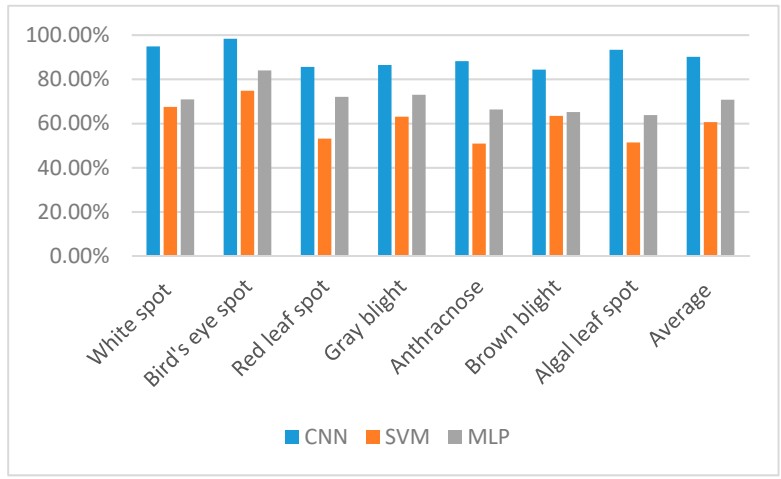

**Figure 5.** Accuracy (as a %) of disease classification for each of the three classification models for the seven candidate tea diseases.

**Table 3.** Error matrix showing the classification accuracy of the LeafNet algorithm.

|  | White Spot | Bird's Eye Spot | Red Leaf Spot | Gray Blight | Anthracnose | Brown Blight | Algal Leaf Spot | Sensitivity | Accuracy | MCA |
|---|---|---|---|---|---|---|---|---|---|---|
| White spot | 111 | 3 | 0 | 0 | 3 | 0 | 0 | 94.87% | | |
| Bird's eye spot | 1 | 117 | 0 | 0 | 0 | 0 | 1 | 98.32% | | |
| Red leaf spot | 0 | 0 | 95 | 7 | 0 | 8 | 1 | 85.59% | | |
| Gray blight | 0 | 0 | 4 | 96 | 3 | 7 | 1 | 86.49% | 90.23% | 90.16% |
| Anthracnose | 5 | 0 | 1 | 6 | 97 | 1 | 0 | 88.18% | | |
| Brown blight | 0 | 1 | 15 | 2 | 0 | 97 | 0 | 84.35% | | |
| Algal leaf spot | 1 | 1 | 2 | 2 | 1 | 0 | 98 | 93.33% | | |

**Table 4.** Error matrix showing the classification accuracy of the SVM algorithm.

|  | White Spot | Bird's Eye Spot | Red Leaf Spot | Gray Blight | Anthracnose | Brown Blight | Algal Leaf Spot | Sensitivity | Accuracy | MCA |
|---|---|---|---|---|---|---|---|---|---|---|
| White spot | 79 | 11 | 0 | 2 | 19 | 1 | 5 | 67.52% | | |
| Bird's eye spot | 12 | 89 | 0 | 4 | 1 | 10 | 3 | 74.79% | | |
| Red leaf spot | 2 | 4 | 59 | 23 | 2 | 19 | 2 | 53.15% | | |
| Gray blight | 0 | 0 | 13 | 70 | 8 | 17 | 3 | 63.06% | 60.91% | 60.62% |
| Anthracnose | 19 | 0 | 5 | 13 | 56 | 11 | 6 | 50.91% | | |
| Brown blight | 0 | 2 | 19 | 17 | 3 | 73 | 1 | 63.48% | | |
| Algal leaf spot | 9 | 10 | 12 | 13 | 3 | 4 | 54 | 51.43% | | |

**Table 5.** Error matrix showing the classification accuracy of the MLP algorithm.

|  | White Spot | Bird's Eye Spot | Red Leaf Spot | Gray Blight | Anthracnose | Brown Blight | Algal Leaf Spot | Sensitivity | Accuracy | MCA |
|---|---|---|---|---|---|---|---|---|---|---|
| White spot | 83 | 13 | 0 | 3 | 15 | 1 | 2 | 70.94% | | |
| Bird's eye spot | 6 | 100 | 0 | 6 | 1 | 5 | 1 | 84.03% | | |
| Red leaf spot | 1 | 1 | 80 | 17 | 0 | 11 | 1 | 72.07% | | |
| Gray blight | 0 | 0 | 9 | 81 | 6 | 14 | 1 | 72.97% | 70.94% | 70.77% |
| Anthracnose | 13 | 0 | 4 | 10 | 73 | 8 | 2 | 66.36% | | |
| Brown blight | 0 | 5 | 16 | 15 | 3 | 75 | 1 | 65.22% | | |
| Algal leaf spot | 6 | 5 | 9 | 10 | 4 | 4 | 67 | 63.81% | | |

**Table 6.** Comparison of LeafNet and two previously published classification algorithms.

| Method | Disease Types Evaluated | Accuracy (%) |
|---|---|---|
| Leaf Net | 7 | 90.16% |
| Algorithm from [26] | 1 | 91% |
| Algorithm from [28] | 2 | 91% |

The disease classification accuracies of the SVM and MLP algorithms were not very high, which is due to the necessity of artificial selection of features. To a large extent, the performance of these methods depends on whether the characteristics selected by investigators are reasonable, while investigators usually rely on experience and can exhibit significant naivety when selecting features. Although better results are obtained by using artificial feature classification, these features are specific for datasets. Results may differ considerably if the same features are used to analyze different data sets, which is a problem inherent to these techniques.

## 4. Conclusions

CNNs have developed into mature techniques that have been increasingly applied in image recognition. The computational complexity needed for neural network analyses is considerably reduced compared to other algorithms and it also significantly improves computing precision. Concomitantly, the high fault tolerance of CNNs allows the use of incomplete or fuzzy background images, thereby effectively enhancing the precision of image recognition.

Feature extraction is an important step in image classification and directly affects classification accuracies. Thus, two feature extraction methods and three classifiers were compared in their abilities to identify seven tea leaf diseases in the present manuscript. These analyses revealed that LeafNet yielded the highest accuracies compared to SVM and MLP classification algorithms. CNNs thus have obvious advantages for identifying tea plant diseases. Importantly, the results from the present study highlight the feasibility of applying CNNs in the identification of tea plant diseases, which would significantly improve disease recognition for tea plant agriculture. Although the disease classification accuracy of the LeafNet was not 100%, improvements upon the present method can be implemented in future studies to improve the method and provide more efficient and accurate guidance for the control of tea plant diseases. At present, the LeafNet model has been applied to the identification of other crop diseases, such as grape disease; however, it needs to be further investigated according to the specific situation of disease, so as to verify the universality of this model.

**Author Contributions:** J.C. contributed to the software, writing and methodology, Q.L. contributed to funding support and revision of this paper, L.G. contributed to the idea and funding support for this paper.

**Funding:** This research was funded by Key R&D Projects of Ningxia Hui Autonomous Region (2017BY080) and National Natural Science Foundation of China (31860477) and the Open Fund of Key Laboratory of Integrated Management of Harmful Crop Vermin in China North-western Oasis, Ministry of Agriculture, China (KFJJ20180108).

**Acknowledgments:** We would like to thank LetPub (www.letpub.com) for providing linguistic assistance during the preparation of this manuscript.

**Conflicts of Interest:** The authors declare no conflict of interest.

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
