# Peer review of "Visual Tea Leaf Disease Recognition Using a Convolutional Neural Network Model"

_symmetry, doi:10.3390/sym11030343_

Round 1
Reviewer 1 Report
In this paper the authors address the problem of tea plant disease classification from leaf images. The manuscript is well written and is very interesting, especially in the part where CNN is treated. The comparison with the other methods is also good in order to give the reader a clear perception of the improvement in performance achieved through deep learning.
In my opinion, the paper needs the following revisions:
- The gold standard for training is not defined as it was obtained; how the target data defining desired network output have been obtained (pathology to each image);
- In Tables 3, 4 and 5 the last column, erroneously referred to as "accuracy", is "sensitivity" (see [1]).
- In this regard it would be advisable to insert the accuracy and the mean class accuracy (MAC).
- It would further enrich the paper a performance comparison table with methods by other authors, even if obtained on different databases
[1] Benammar Elgaaied, "Computer-assisted classification patterns in autoimmune diagnostics: The AIDA project", BioMed Research International 2016 (open access)
Author Response
Point 1: The gold standard
for training is not defined as it was obtained; how the target data
defining desired network output have been obtained (pathology to each
image);
Response 1: The gold standard for training we used according to the previous study [14] and [33].We obtained the target data through the plant pathologist identification and pathogen identification by morphology and molecular methods, and the identification criteria was referred to [35] and [36].
Point 2: In Tables 3, 4 and 5 the last column, erroneously referred to as "accuracy", is "sensitivity" (see [1]).
Response 2: Revised accordingly.
Point 3: In this regard it would be advisable to insert the accuracy and the mean class accuracy (MAC).
Response 3: Revised accordingly. We have inserted the accuracy and the mean class accuracy in the text.
Point 4: It would further enrich the paper a performance comparison table with methods by other authors, even if obtained on different databases
Response 4: We made a comparison between the LeafNet model and the references [26] and references [28], the result was presented in Table 6.
Reviewer 2 Report
The paper proposes approaches for tea leaf disease recognition using a revised version of AlexNet, plus with two additional approaches using MLP and SVM based on bag of words visual features. It is a good study. Few important consideration should be paid:
There needs to have a clear explanation how the proposed LeafNet differs from AlexNet.
Why do you choose AlexNet as the base machine learning model for tea leaf recognition? There are so many models out there and it is important that the authors clearly discuss this consideration.
There are recent literature on using neural network based method for agriculture engineering. Some of good examples are indicated below:
“Computational Deep Intelligence Vision Sensing for Nutrient Content Estimation in Agricultural Automation,” IEEE Trans. Automation Science and Engineering, vol. 15, no. 3, pp. 1243-1257, 2018
“Building A Globally Optimized Computational Intelligence Image Processing Algorithm for On-Site Nitrogen Status Analysis in Plants,” IEEE Intelligent Systems, vol. 33, no. 3, pp. 15-26, 2018
"Regularized Neural Networks Fusion and Genetic Algorithm based On-Field Nitrogen Status Estimation of Wheat Plants,” IEEE Trans. On Industrial Informatics, vol. 13, no. 1, pp. 103-114, 2016
These work also uses similar method and the authors should have a good discussion about them in the literature section.
For the completeness of information, a discussion should be allocated on why the SIFT and BAg Of Words are used.
There are many typo in the manuscript. For example, in Figure 4, "clustring" should be spelt as clustering.
Author Response
Point 1: There needs to have a clear explanation how the proposed LeafNet differs from AlexNet.
Response 1: The number of filters in the first, second, and fifth convolutional layers are designed to equal half of those used in AlexNet's filters. Further, the number of neurons in the fully connected layer are 500, 100, and seven, respectively, which differ from the number used in the standard AlexNet architecture.
Point 2: Why do you choose AlexNet as the base machine learning model for tea leaf recognition? There are so many models out there and it is important that the authors clearly discuss this consideration.
Response 2: Due to the number of parameters and layers of AlexNet is moderate than the other machine learning models, it is easier to achieve. The total number of parameters (weights and biases) of the entire AlexNet network reaches upwards of 60 million, wherein the parameters of the convolution layer comprises 3.8% of the total network parameters and those of the full connection layer comprise up to 96.2% of the total. To reduce the computational complexity associated with such networks, we sought to construct a disease identification model with a relatively simple structure and small computational requirements. The network was constructed by reducing the number of convolutional layer filters and the number of nodes in the fully connected layer, thereby effectively reducing the number of network parameters requiring training and reducing the overfitting problem.
Point 3: These work also uses similar method and the authors should have a good discussion about them in the literature section.
Response 3: Revised accordingly. We have a discussion about them in the literature section.
Point 4: For the completeness of information, a discussion should be allocated on why the SIFT and Bag of Words are used.
Response 4: The HOG, SURF, and PHOW algorithms did not exhibit better invariance towards image rotation and scaling than SIFT, while the SIFT algorithm performed reliably with affine transformations, viewing angle variation, and noise. Moreover, the SIFT algorithm exhibited strong scalability, that when combined with other algorithms could be used as a highly optimized algorithm. Consequently, SIFT was used here as a feature descriptor in a traditional machine learning algorithm. Although SIFT features can describe images, each SIFT represents a 128-dimensional vector, and images contain hundreds or thousands of SIFT features, thereby leading to very computationally intensive operations. To greatly reduce computational processing, a Bag of Visual Words (BOW) Model was constructed based on these vectors, wherein each image was represented by a numerical vector.
Point 5: There are many typo in the manuscript. For example, in Figure 4, "clustring" should be spelt as clustering.
Response 5: Revised accordingly.
Reviewer 3 Report
The authors proposed a method for identifying tea leaf diseases using LeafNet and compared its accuracy with SVM and MLP. The paper is generally well written and easy to follow. There are some minor issues that must be solved:
In line 55 instead of "sigmoidal/linear activation functions" it is written "sigmoidal linear activation functions";
The sentence started in line 57 ("SVMs are...") must be reshaped;
The sentence started in line 76 ("Meanwhile CNNs...") must be reshaped;
The sentence started in line 167 ("Layer parameters...") must be reshaped;
The sentence started in line 201 ("The final step...") must be reshaped;
Line 208: a space must be inserted between "1.0" and "The";
Author Response
Point 1: In line 55 instead of "sigmoidal/linear activation functions" it is written "sigmoidal linear activation functions";
Response 1: The sigmoidal linear activation functions was used to map the sum of the weighted inputs to the output of a neuron in the hidden layer.
Point 2: The sentence started in line 57 ("SVMs are...") must be reshaped;
Response 2: SVMs are an effective type of classification algorithm that has been widely used for many pattern recognition tasks, including machine learning methods based on statistical learning theory.
Point 3: The sentence started in line 76 ("Meanwhile CNNs...") must be reshaped;
Response 3: Meanwhile, the high recognition accuracy of CNNs lead to wide implementation in fields related to computer vision, where development is occurring rapidly.
Point 4: The sentence started in line 167 ("Layer parameters...") must be reshaped;
Response 4: Layer parameters for the LeafNet are shown in Table 2.
Point 5: The sentence started in line 201 ("The final step...") must be reshaped;
Response 5: The final step is the operation of a classifier, which was either a support vector machine (SVM) or Multi-Layer Perceptron (MLP) algorithm in this study.
Point 6: Line 208: a space must be inserted between "1.0" and "The";
Response 6: Revised accordingly.